# Self-organizing maps of typhoon tracks allow for flood forecasts up to two days in advance

Li-Chiu Chang[1] ✉, Fi-John Chang[2] ✉, Shun-Nien Yang[1], Fong-He Tsai[2], Ting-Hua Chang[3] & Edwin E. Herricks[4]

Typhoons are among the greatest natural hazards along East Asian coasts. Typhoon-related precipitation can produce flooding that is often only predictable a few hours in advance. Here, we present a machine-learning method comparing projected typhoon tracks with past trajectories, then using the information to predict flood hydrographs for a watershed on Taiwan. The hydrographs provide early warning of possible flooding prior to typhoon landfall, and then real-time updates of expected flooding along the typhoon's path. The method associates different types of typhoon tracks with landscape topography and runoff data to estimate the water inflow into a reservoir, allowing prediction of flood hydrographs up to two days in advance with continual updates. Modelling involves identifying typhoon track vectors, clustering vectors using a self-organizing map, extracting flow characteristic curves, and predicting flood hydrographs. This machine learning approach can significantly improve existing flood warning systems and provide early warnings to reservoir management.

[1] Department of Water Resources and Environmental Engineering, Tamkang University, New Taipei City 25137, Taiwan. [2] Department of Bioenvironmental Systems Engineering, National Taiwan University, Taipei 10617, Taiwan. [3] The Fifth River Management Office, Water Resources Agency (WRA), Ministry of Economic Affairs, Taipei, Taiwan. [4] Department of Civil and Environmental Engineering, University of Illinois at Urbana-Champaign, Urbana, IL 61801-2352, USA. ✉email: changlc@mail.tku.edu.tw; changfj@ntu.edu.tw

aiwan's location makes it vulnerable to the western North Pacific Ocean tropical cyclone that regularly produces damaging typhoons. Although small in size, Taiwan is 400 km long and 150 km wide, mountains reach elevations of 4000 m. With a northeast to southwest orientation the mountains dominate the eastern part of the island. Rivers in Taiwan are characterized as steep gradient and short length producing high flows within a few hours of typhoon passage. Existing reservoirs, in general, are small and are rapidly filled during typhoon-related rainfall resulting in significant flood hazards[1,2]. For example, the passage of Typhoon Morakot had major flooding produced by 2777 mm of rainfall[3].

The increase in the number of unprecedented weather events, including typhoons, is a part of weather patterns related to global temperature change[4,5]. In Taiwan, global temperature change is expected to increase the frequency of damaging typhoons[6–8]. When typhoons cross landscapes with mountainous terrains, such as Taiwan, intense localized rainfall and flooding is common due to orographic effects. Flood damage potential is related to typhoon intensity class, the track each typhoon takes across the island, and orographic influences of topography, which influence local rainfall amount and intensity. To minimize typhoon-related flood damage there is a clear need for improved flood forecasting with early warnings that provide sufficient time to implement flood hazard mitigation using reservoir storage and local flood defense.

Accurate advanced forecasting of flooding is a formidable challenge in Taiwan. Variable typhoon tracks, trajectory, speed, and rainfall intensity related to mountainous terrain produce spatial and temporal variability in rainfall amounts and related runoff[9,10]. Predictive model advancements are needed to provide the lead time needed to adjust reservoir capacity for flood control and implement flood defense procedures while meeting long-term water supply requirements. Improved sensor networks in Taiwan now provide easily accessible remote sensing data, expanding modelling potentials[11,12]. A recent study of rainfall-runoff modeling based on remote rainfall information found that reliable real-time flood forecasts could be obtained up to six hours before a typhoon event[3]. This six-hour forecast falls well short of the time needed for reservoir storage development, which is measured in days. Therefore, it is important to improve traditional modeling and rainfall-runoff analyses to develop typhoon track and flood hydrograph predictions several days before typhoon landfall.

Prediction of flood hydrographs is based on the spatio-temporal variability of storm characteristics and the uncertainty of hydro-geo-meteorological outcomes along a track produced by track/terrain interactions. Artificial intelligence (AI) techniques have recently emerged as an approach to analyzing highly dimensional complex data sets to classify phenomena and support predictions[13–21]. The AI techniques analyzed geographic, hydrological, and meteorological data sets to enable automatic information extraction from advanced sensor arrays through machine learning[22–28]. There is ample evidence that the use of AI-based approaches has improved site-specific rainfall-runoff prediction for individual typhoons, bridging the gap between track prediction and flood forecasting by combining the analysis of massive historical datasets and real-time remote sensing data.

The AI-based methodology developed in this study predicts flood hydrographs based on the forecasts of a typhoon track both before and after typhoon landfall and then constantly refines flood forecasts during typhoon passage. Our study made the first attempt to digitize analog typhoon tracks so that the whole typhoon track, together with corresponding hydrologic and geographic characteristics, could be integrated to support hydrograph prediction. We based predictions on clusters of vectorized historic typhoon tracks using a self-organizing map (SOM) with event-specific flow characteristic curves (FCCs), which are based on reservoir measurements related to typhoon landfall, overland transit expectations and rainfall amount. Our study showed that it was possible to make predictions with a lead time of two days prior to typhoon landfall. We could provide an early warning coupled with continuous prediction updates of flooding along the typhoon track. Better predictions improve reservoir operation, flood defense and integrated water resources management.

## Results

**Reservoir watershed in need of flood hydrograph prediction.** In this study, the Shihmen Reservoir watershed was the focus for flood forecasting. This watershed experiences an average of three typhoons annually. The Shihmen Reservoir is a pivotal multi-objective reservoir that provides flood protection and a water supply of more than 800 million m³/year to meet the domestic, agricultural, and industrial needs of the Taipei metropolitan area, thus requiring careful management. The reservoir has a capacity of 197 million m³ with a watershed area of 763 km². The annual precipitation for the Shihmen Reservoir watershed is about 2500 mm, primarily from typhoon-related rainfall. Reservoir operational rules require reduction in the water level before typhoon arrival to increase storage capacity. Storage is an element of flood defense that is balanced against maintaining a high water level to meet water supply needs. Therefore, reliable and accurate flood hydrograph prediction during typhoon periods is crucial for reservoir management to provide flood defense while meeting water supply requirements.

**Approach to predicting typhoon-induced flood hydrographs.** Typhoon track predictions are possible by analyzing historic tracks. Similar tracks can be grouped, and groups can be classified based on the path over Taiwan using K-means and fuzzy clustering as prediction tools[29–34], but these methods must be improved to provide digital representations of typhoon tracks to support the location-specific analysis for predicting rainfall intensity related to terrain. Improvements needed include analog to digital conversion of typhoon movement into digital track vectors, characterization of track variability, and accounting for track interactions with terrain.

We assembled hydro-meteorological data from 97 typhoons occurring between 1965 and 2019 that tracked through the Shihmen Reservoir watershed producing flood hydrographs exceeding 600 cms. The maximum flow from the reservoir during this period was 8594 cms. The data set supporting the AI-based flood hydrograph prediction model developed in this research used total rainfall, typhoon track, the date and time warnings issued by the Central Weather Bureau (CWB), and hourly reservoir inflow data from the Water Resources Agency (WRA). The 97 typhoon events were divided into two groups, where 87 events were used to train the model and 10 events were used to test model reliability.

Prediction of typhoon-induced flood hydrographs involved the sequential implementation of four key modules, which were typhoon track vectorization, track vector clustering in a SOM, FCC extraction, and flood hydrograph prediction.

**Typhoon track vectorization.** We first converted each typhoon track from an analog to a digital data set (vector). Recognizing potential track variability, a diffusion process was applied to better characterizing each track. Figure 1 illustrates the vectorization of a typhoon track passing across Taiwan with results of the diffusion process presented. This track vectorization method

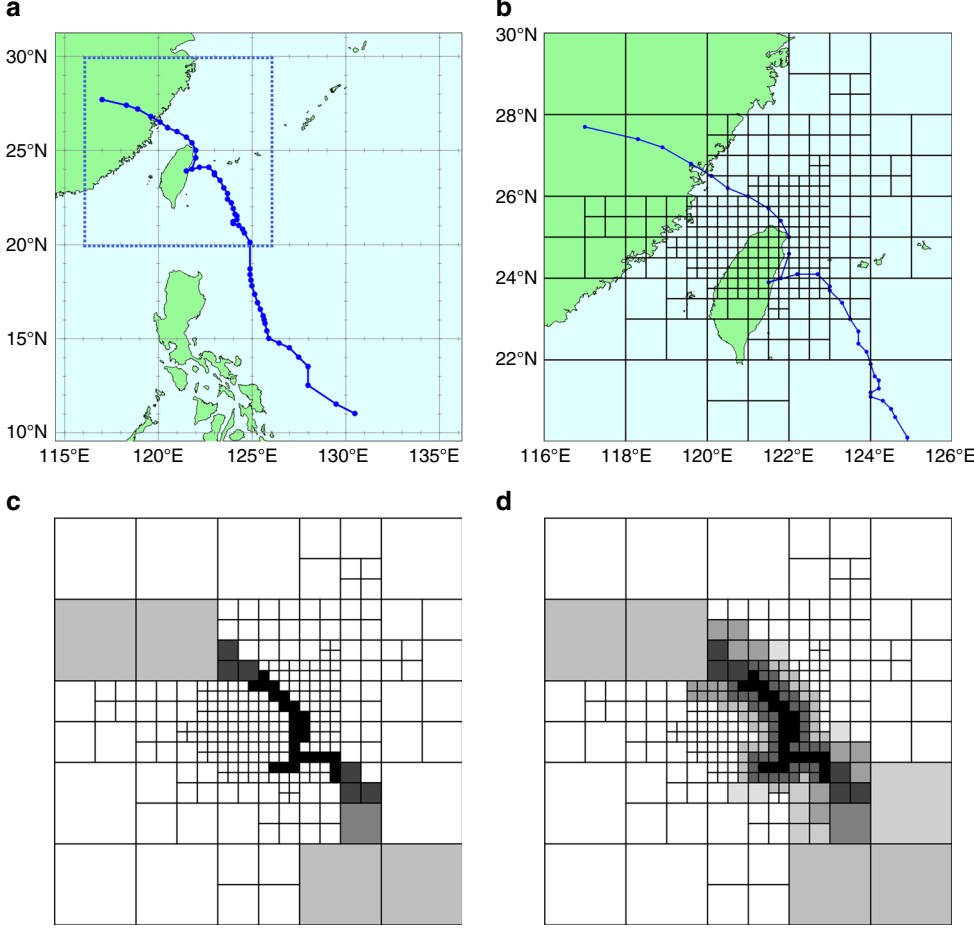

**Fig. 1 Vectorization process of a typhoon track passing across Taiwan. a** An example typhoon track passing over Taiwan. **b** An example typhoon track with the grid used to vectorize tracks. Variable size grid cells address topographic conditions (see Methods). **c** Each grid cell through which the typhoon moves is shaded. **d** To assess the spatial relationships of typhoon impacts, a weight diffusion process is applied to track plotting so that grid cells close to the typhoon track is also included, but with lower weights. The shade of each grid cell represents its weight (see Methods).

allowed a decomposition of a continuous (analog) track into a discontinuous grid vector without losing key features (e.g., direction, speed, and duration). We produced digitized typhoon tracks for all the 97 events using the vectorization procedure on 277 grids.

**Track vector clustering in a SOM.** The second module used a SOM algorithm to cluster the 87 tracks occurring between 1965 and 2015 selected for model training into a topological display. The 87 tracks are illustrated in the Supplementary Fig. 1. Figure 2 shows the $4 \times 4$ SOM results where vectorized typhoon tracks similar in shape are grouped into the same cluster (neuron), and Fig. 3 displays individual typhoon tracks classified in their own cluster, where different typhoon tracks in each cluster are presented in different colors. Typhoons moved across different areas of the Shihmen Reservoir watershed producing different rainfall characteristics. The clustering results grouped similar tracks in the same cluster. It was noted that the tracks behaved more consistently between adjacent clusters than non-adjacent clusters. For example, the typhoons approaching the northern coast where rainfall occurred in the reservoir watershed clearly influenced reservoir inflow (e.g., typhoon tracks in clusters #9, #10, #13, and #14). This contrasts with typhoons that missed the Shihmen Reservoir watershed and only low inflow was produced (e.g., typhoon tracks in clusters #7, #8, and #12).

Typhoon tracks in clusters #1–#7 came mainly from southeast, made landfall on the east coast, and moved across the 4000 m high mountains. In typhoons with this east to west movement, the orographic processes caused rain to fall in other watersheds while producing lower inflows to the Shihmen Reservoir. As for typhoon tracks in clusters #11 and #15, these tracks were east or north of the watershed, again resulting in lower reservoir inflows. According to the 16 clusters, the cluster that a new typhoon track best matched could be quickly and objectively identified.

**FCC extraction.** FCC is a cumulative curve that shows the percent of time specified discharges were equaled or exceeded during a typhoon event. It has a long history in the field of water resource engineering and for scientific comparisons of streamflow characteristics across watersheds. The normalized flow characteristic curve plots the fraction of total discharge in the vertical axis against the fraction of duration in the horizontal axis. The third module produced FCCs, where the primary indicator used was the slope for the rising and recession limbs of the hydrograph. Different watersheds have site-specific flow characteristics and duration time for the FCC. For the Shihmen Reservoir watershed, typhoon effect on inflow would last for an average of 28 h with an average time interval between the arrival and departure of a typhoon reaching 56 h (based on 87 typhoon events). To provide reservoir capacity for flood control the change from 245 m (full reservoir storage) to 240 m (the upper limit)

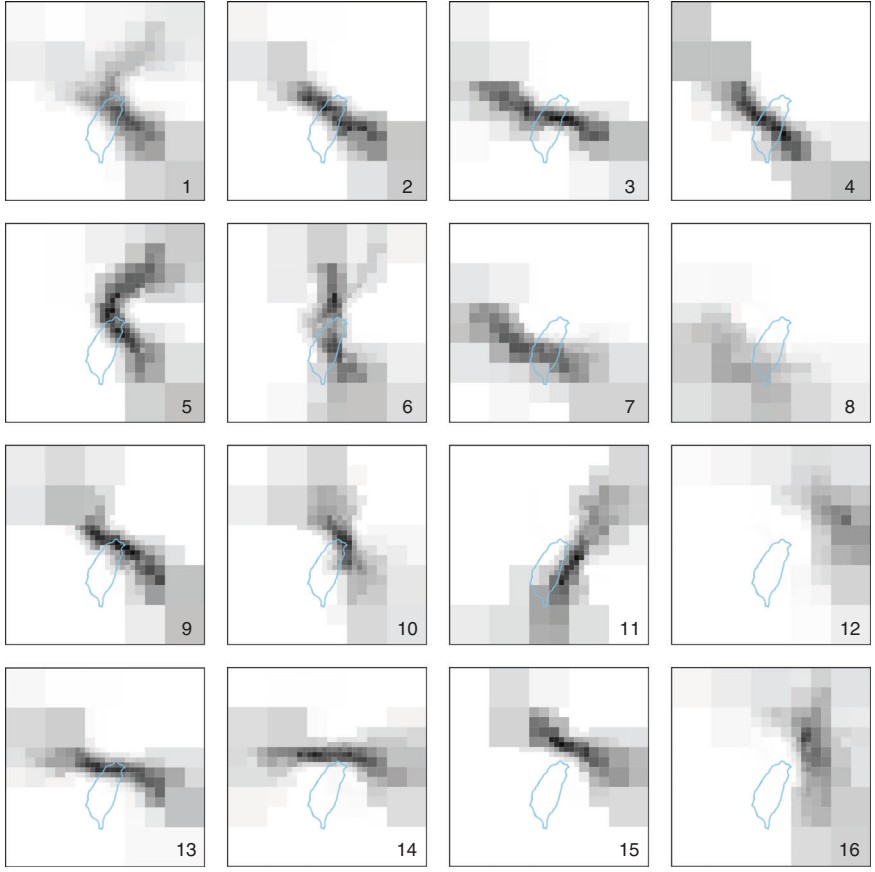

**Fig. 2 Results of track vectorization of 87 typhoons.** Vectorized typhoon tracks similar in shape are grouped in clusters based on the 4 × 4 self-organizing map. These subfigures include weight shading. Clusters are numbered, as shown in the bottom right corner of each cluster.

would take about 11 h at a rate of 1000 cms or about 37 h at a rate of 300 cms. Therefore, the prediction of flood hydrograph with a lead time of two days prior to typhoon landfall would be satisfactory for flood control as well as water management.

Hydrographs were predicted from rainfall forecasts and typhoon duration (TD). In this study, the FCC of a typhoon event was normalized by a cumulative flow curve where both the total flow rate and TD were converted to the same 0 to 1 scale. It is understood that terrain complexity and flow travel distance produced by each typhoon track brings different effects on the watershed. We argue that similar typhoon tracks would produce similar effects reducing analytical complexity. FCCs in the same cluster could be estimated by use of the total flow volume that was converted directly from total rainfall. The advantage of this approach is that the calculation of flow hydrographs no longer requires actual rainfall–runoff data, which is only available post-event. Further, using this approach a flood hydrograph could be converted from a FCC cluster and an expected typhoon track supporting the development of advanced flood warnings.

We investigated the characteristics of the flood hydrographs in each cluster, which was initiated with an extraction of the FCC for each typhoon event. We then analyzed the features (e.g., duration, peak flow, and time required to reach peak flow) and similarity of all the FCCs in each cluster. Figure 4 shows the FCCs with time required to reach peak flow (TP, hour), peak flow (QP, cms), and (TD, hour) for each cluster. We found that the shape of an FCC primarily depended on typhoon stage/duration. Considering this time dependence, we used three schemes to characterize TD. The first scheme characterizing TD spans between the arrival and departure time of a typhoon over the gridded zone (Fig. 4). The second scheme spans between the start rising flow limb

characteristics and the departure of the typhoon from the gridded zone (Fig. 5). The third scheme spans between the start rising limb characteristics and the cessation of rainfall (Fig. 6).

This approach allowed us to explore how typhoon stage/duration would influence the group behavior of FCCs in each cluster (Figs. 4–6). We noticed from Fig. 4 that the shape of the FCC was not affected by typhoon scale or intensity and similar FCCs were scattered in most of the clusters while the TD was identifiable only in clusters with a minimum duration of 48 h and a maximum duration of 167 h (cluster #5). We also noticed that after a typhoon entered the grid (about 600 km from Taiwan) the early stages of the rising limbs were inconsistent in each cluster. This inconsistency was also noticed in recession limbs (i.e., associated with typhoon departure from the gridded zone). To explore the cause of this inconsistency, TD was re-defined using the duration between the time that flow significantly increased and the time that either the typhoon moved away from the watershed (Fig. 5) or rainfall stopped over the watershed (Fig. 6). The result of this redefinition was a shorter TD. The FCCs in each cluster were then more similar in shape when effective TD was reduced. For example, the difference between start-end hours in neuron #5 was reduced from 48–167 (Fig. 4) to 35–120 (Fig. 5), and to 23–66 (Fig. 6), respectively. This supported the notion that a close relationship between typhoon tracks and their corresponding flood hydrographs could be obtained in each cluster when our TD calculation methods were implemented.

A storm may approach the watershed from various directions with different rainfall histograms (patterns), causing the timing of flow rising to peak (TP) to be variable. As shown in Fig. 4, TPs of the 87 typhoon events ranged between 7 and 55 h, but the mean TP values in the 16 clusters tended to be less than 24 h, which

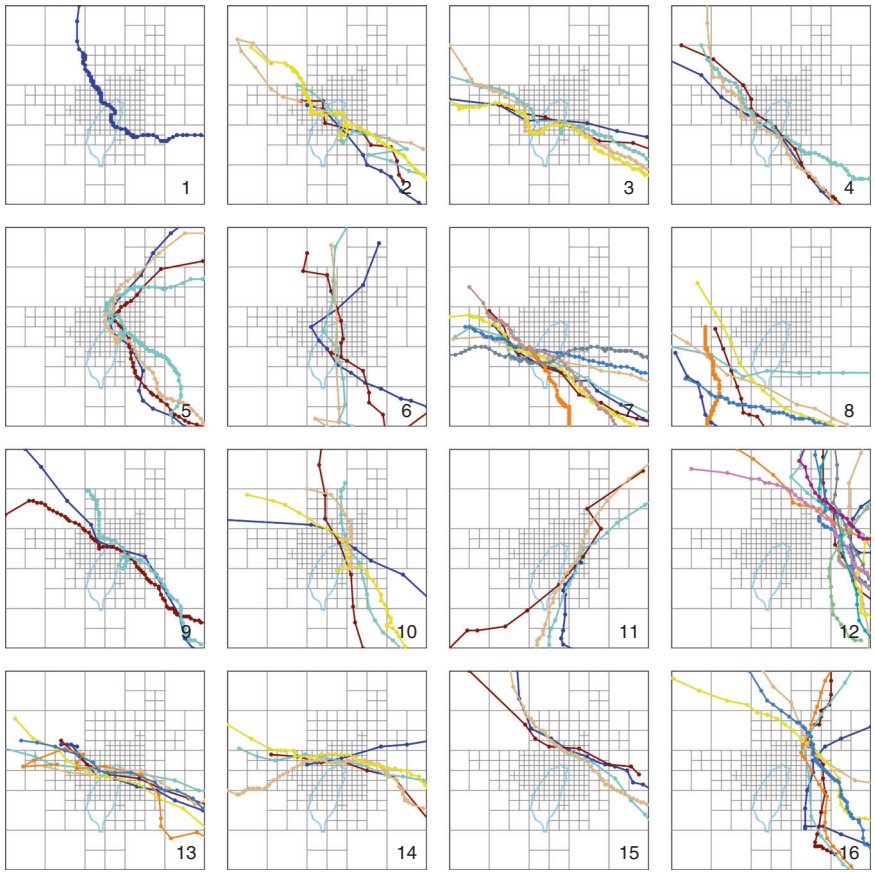

**Fig. 3 Individual typhoon tracks clustered in the 4 × 4 self-organizing map.** Each typhon track in a cluster is plotted over grid cells. Different colors plot different typhoon tracks.

made the reservoir operation difficult because of the inherent lag time between alert and achieving the desired capacity was insufficient. According to typhoon data of this study, peak flow (QP) varied in a range from 662 cms to 8594 cms. The highest QP values in clusters #1–#8 were generally <3000 cms while all the highest QP values in neurons #9, #10, #13, and #14 exceeded 5000 cms. The configured SOM topological map showed that the QP values were in general much smaller in the first two rows than in the last two rows. This shows a clear indication of flow-related impacts related to typhoon track, providing a useful tool to suggest crucial guidance for flood defense and management. The utility of TP, QP, and TD in defining typhoon characteristics related to flooding provides a key to exploring the hydrograph predictor.

**Flood hydrograph prediction.** Given the predicted track of an approaching typhoon, a flood hydrograph prediction process began by identifying from the SOM topological map a cluster (denoted as the best matched cluster) that incorporated a typhoon track the most similar to the predicted track. The FCC prediction could use either of the following two selection strategies. The first strategy selected the FCC of the best matched typhoon track while the second strategy selected the average (ensemble) of all FCCs in this best matched cluster. Using either approach, the prediction of the flood hydrograph of an approaching typhoon could be made and flood warnings could be generated. To summarize model training, we recognize that past methods for predicting typhoon impacts can benefit from improved methods of typhoon track prediction[28,35], and with these analyses impact prediction can be further improved with flood forecasting using our AI approach.

We next evaluated the reliability of the constructed SOM embedded with FCCs using the remaining 10 typhoon events occurring in 2013 and 2019 as tests of our approach. The test results for 10 typhoon events are shown in the Supplementary Table 1. We found that the predicted flood hydrographs of the 10 test events generally matched the actual flood hydrographs providing a long lead time (e.g., several days), with acceptable variation in the timing and volume of peak flows. This is a major improvement of existing prediction modeling approaches (e.g., physical, conceptual, and data-driven) that focus on rainfall-runoff mechanisms providing a short lead time (e.g., one- to six-hour)[36–38]. Figures 7–9 present the predicted flood hydrographs of the three test typhoon events (i.e., Typhoon Fitow, Typhoon Soulik, and Typhoon Dujuan, respectively) under three TD schemes implemented with the two FCC selection strategies. The results provided in Fig. 7 had classified Typhoon Fitow into cluster #15 (Figs. 2 and 3). When implemented with the first FCC selection strategy (the best matched track, Fig. 7b), the flood hydrograph of Typhoon Fitow could be predicted almost perfectly based on the FCC of the best matched track (Typhoon Cora) under the second (Fig. 5) and the third (Fig. 6) TD schemes. When implemented with the second FCC selection strategy (average (ensemble) of all FCCs in the best matched cluster, Fig. 7c), the flood hydrograph of Typhoon Fitow could also be predicted nicely by averaging the four FCCs in the best matched cluster (#15) under the third TD scheme (Fig. 6). As for Typhoon Soulik shown in Fig. 8, it was classified into cluster #13 (Figs. 2, 3). When implemented with both FCC selection strategies (Fig. 8b, c), the flood hydrograph of Typhoon Soulik could be suitably predicted by the FCC of the best matched track (Typhoon Herb) and by the average of the seven FCCs in the best

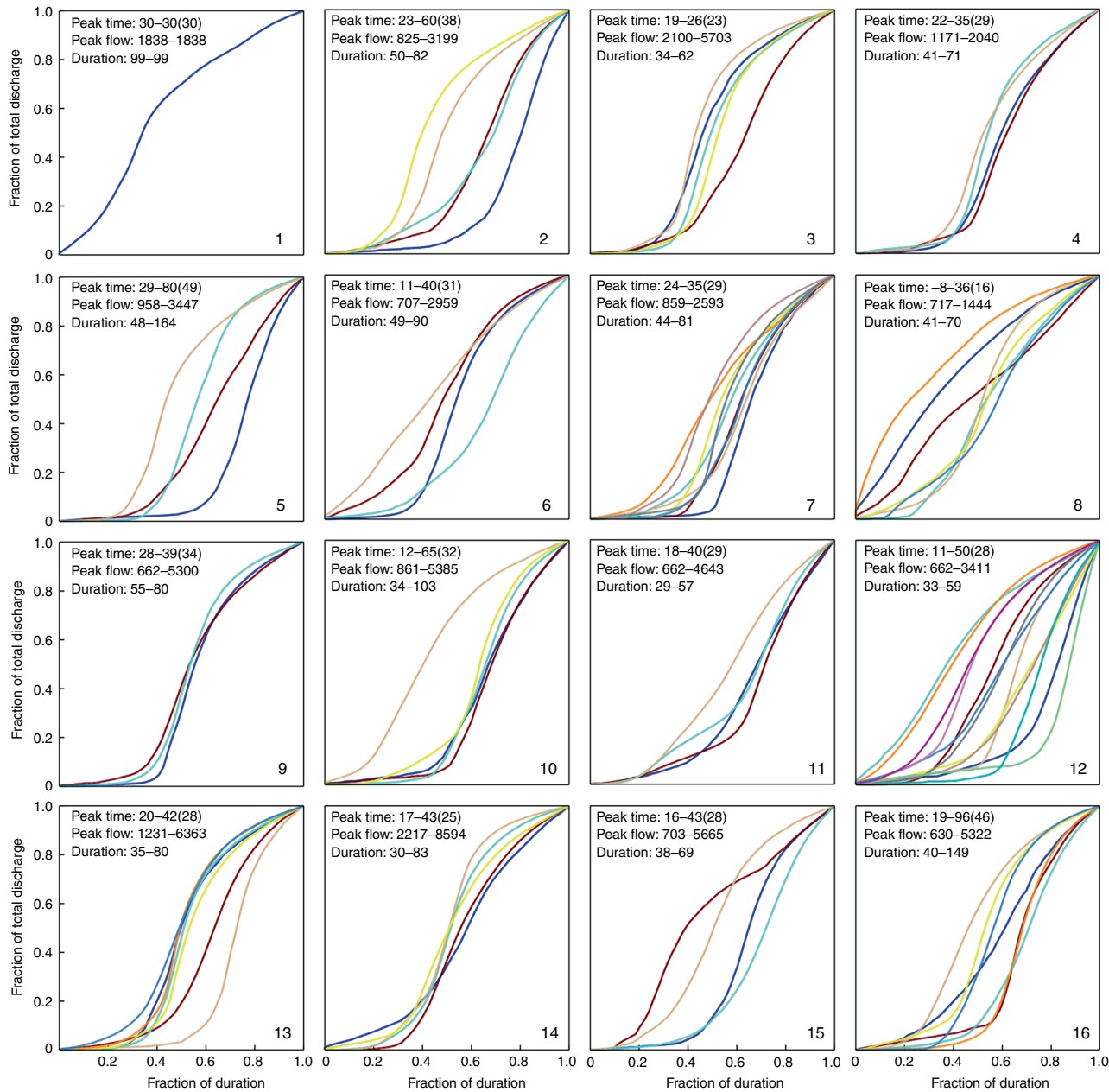

**Fig. 4 Normalized flow characteristic curves using the first typhoon duration scheme.** The first scheme determining typhoon duration spans between the arrival and departure time of a typhoon over the gridded zone. A normalized flow characteristic curve plots the fraction of total discharge against the fraction of duration. Different colors are shown for each typhoon track.

matched cluster (#13) under the third TD scheme (Fig. 6). Regarding Typhoon Dujuan shown in Fig. 9, it was classified into cluster #3 (Figs. 2, 3). When implemented with the first FCC selection strategy (Fig. 9b), the flood hydrograph of Typhoon Dujuan could be perfectly predicted based on the FCC of the best matched track (Typhoon Talim) under the second (Fig. 5) and the third (Fig. 6) TD schemes. When implemented with second FCC selection strategy (Fig. 9c), the flood hydrograph of Typhoon Dujuan could also be well predicted by the average of the five FCCs in the best matched cluster (#3) under the third TD scheme (Fig. 6).

### Discussion

Because of variation in typhoon intensity and track is influenced by landscapes in Taiwan, flash flooding is common within few hours of typhoon passage, and reservoirs quickly fill during

typhoon events. Using hydrographs for reservoir capacity management is a crucial non-structural approach to flood defense in water resources management. Modeling the rainfall–runoff processes is one of the most popular yet complex practices of hydro-informatics approaches (e.g., conceptual, physical, machine learning models) while the high degree of spatio-temporal heterogeneity of typhoon-induced rainfall and the notoriously nonlinear nature of rainfall–runoff relationship make reliable and accurate flood forecasts very challenging, if not impossible. Moreover, traditional approaches now only make short-term (hourly-based in our case) forecasts due to the lack of reliable rainfall predictions. As noted in the literature, forecast accuracy deteriorates as the lead time increases[39]. These constraints restrict operational hydrology applications in watershed management.

To translate short-term (hourly-based) flood forecasting into long-term (daily-based) flood warning using typhoon tracks, we

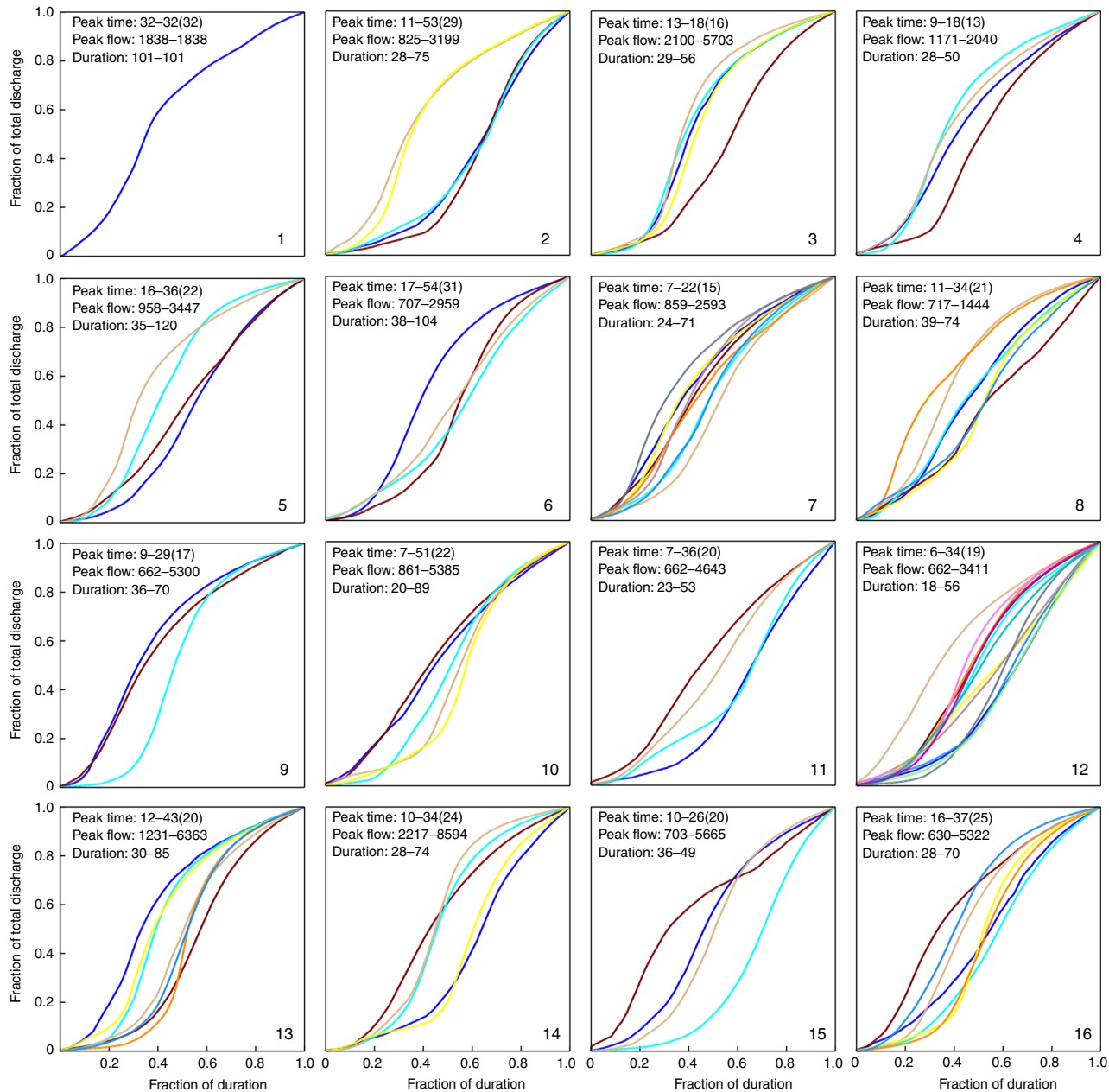

**Fig. 5 Normalized flow characteristic curves using the second typhoon duration scheme.** The second scheme determining typhoon duration spans between the start rising flow limb characteristics and the departure of the typhoon from the gridded zone.

adopted the AI-based approach to predict flood hydrograph based on the forecasted typhoon tracks and corresponding total rainfall obtained from our CWB. To assess the reliability and accuracy of this AI-based approach, we compared the prediction results with those of a commonly used conceptual rainfall-runoff model, i.e., the storage function model (SFM)[40,41].

As presently developed, the SFM cannot be applied without known rainfall patterns. Thus, the historical rainfall patterns and the corresponding runoff hydrographs of 97 typhoon events were used for SFM modeling. In this modeling 87 events were used for training and the remaining 10 events for testing. We noticed that because only rainfall patterns were available, the results could only be treated as simulated rainfall-runoff patterns, rather than predicted runoff based on the previous rainfall histogram. Therefore, the simulation results were adopted only to assess the goodness-of-fit of our AI-based approach for predicting flood hydrographs. The results of the AI-based and SFM methods for

the ten testing events are summarized in the Supplementary Table 1. It appears that the proposed AI-based method improved the performance for all the test events, in terms of smaller RMS and larger $R^2$ values, especially apparent for events with high peaks. To demonstrate the goodness-of-fit of both methods, we examined performance for two special typhoon events, i.e., the most recent typhoon in 2019, Typhoon Lekima (Supplementary Fig. 2), and the highest flood hydrograph, Typhoon Aere in 2014 (Supplementary Fig. 3). Results show that the AI-based approach is superior to the SFM method. This is especially true for Typhoon Aere, whose peak is the highest among those of 97 events. The results show that the AI-based approach with the best match strategy could fit the historical flood hydrograph very well while the SFM method, which has three parameters calibrated based on training datasets, could not produce a suitable flood hydrograph and significantly underestimated the peak. Consequently, we conclude the AI-based approach can, in general,

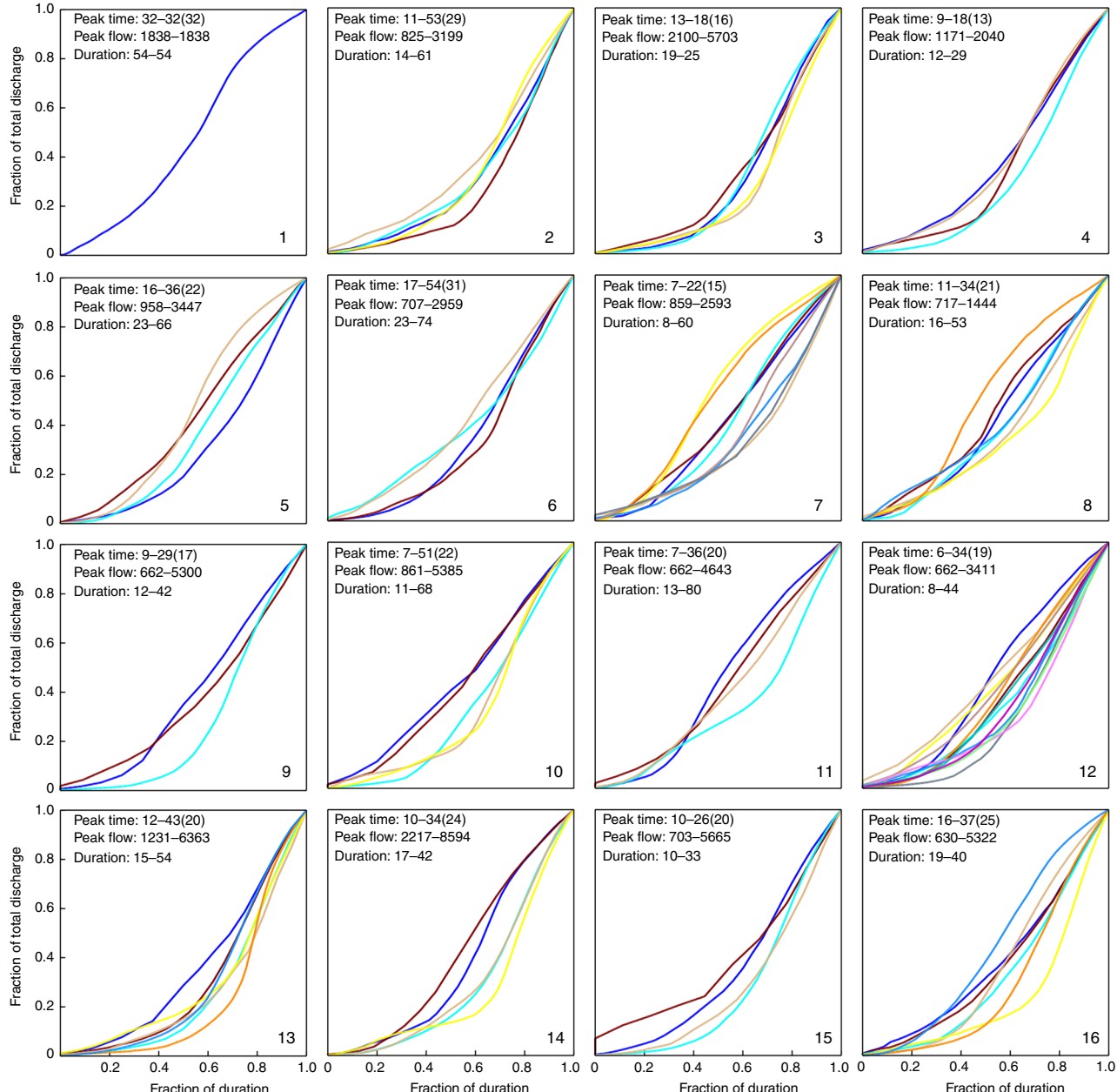

**Fig. 6 Normalized flow characteristic curves using the third typhoon duration scheme.** The third scheme determining typhoon duration spans between the start rising limb characteristics and the cessation of rainfall.

obtain reliable and accurate prediction of flood hydrographs based on known typhoon tracks and corresponding total rainfall amounts. Furthermore, we notice that the strategy based on the best matched typhoon track, as expected, could, in general, produce more favorable results than those based on the averaged ensemble tracks. Thus, the accuracy of typhoon track prediction is one of the most crucial factors affecting flood hydrograph prediction.

It is not uncommon to incorrectly predict typhoon track, consequently the track would be associated into different clusters and result in poor flood forecasting. As known, the SOM is a powerful tool to form a two-dimensional topological map where similar tracks would be placed in the same cluster and relative tracks would be placed in the adjacent clusters. The benefit of this track clustering was the ability to match an approaching typhoon track with a typhoon track similar in shape present in the SOM. Nevertheless, in case of incorrect track forecast (but within

certain range), it would be classified into neighboring clusters and produce relatively good predictions. Thus, the proposed method could tolerate small errors in typhoon track prediction, which provides a robust error-tolerant approach. We present a recent case, Typhoon Lekima in 2019, which had a poorly predicted track, with an error of 80 km away from the north of Taiwan. The original predicted track should be mapped into cluster #15, while the actual track is found in cluster #16. The projected results of the actual track and the original predicted track are given in Supplementary Figs. 4 and 5, respectively. We notice that inaccurate track prediction does cause some differences in both the forecasted flood peak and the occurrence timing of a peak while these differences fall within a small (acceptable in a management context) range. Thus, the SOM clustering method deals with false predictions in an error-tolerant manner.

We notice that the quality of tropical cyclone (TC) track forecasts, especially for the track of the TC's center, has been

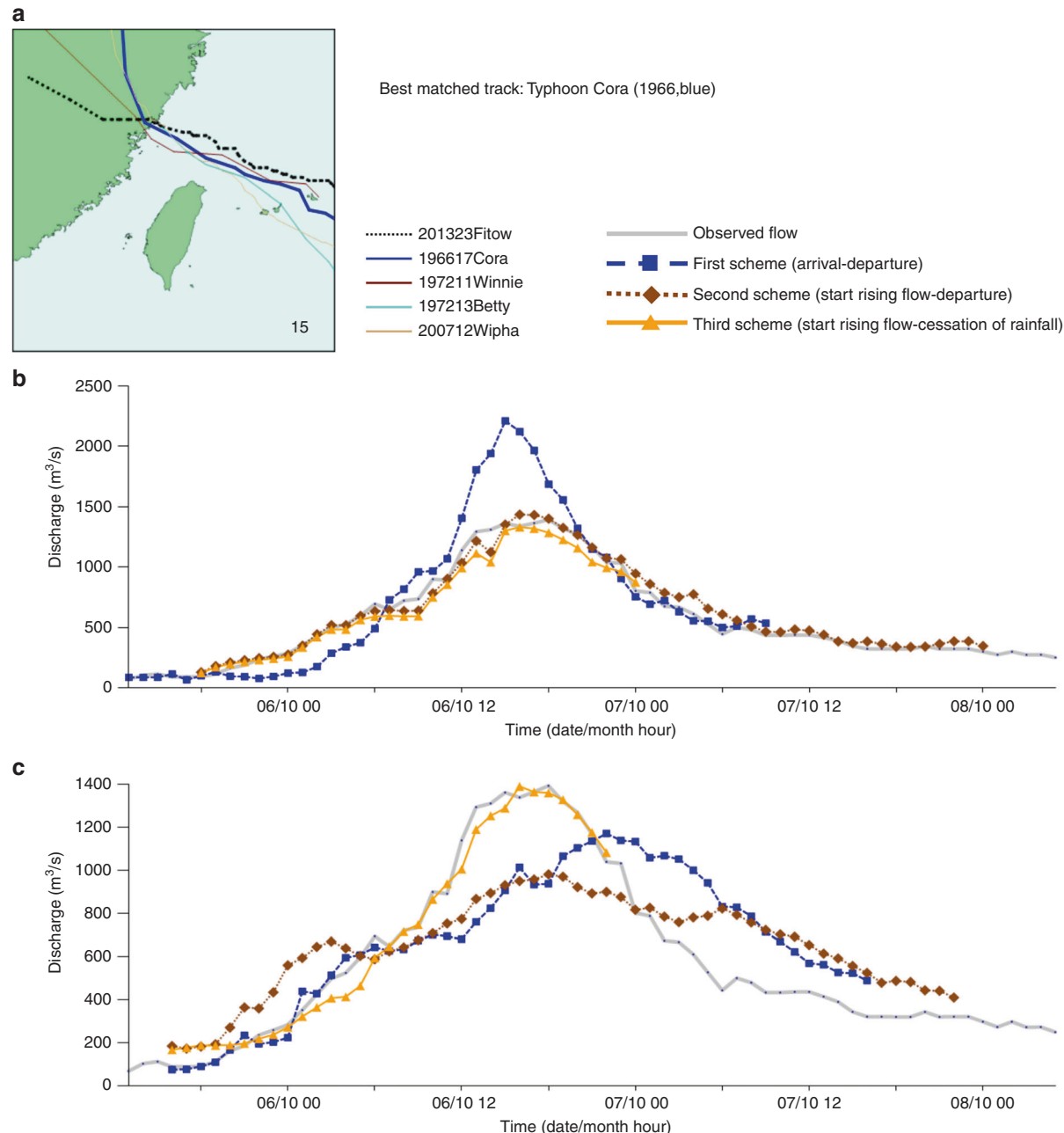

**Fig. 7 Predicted flood hydrographs for Typhoon Fitow.** Predictions were made using two flow characteristic curve selection strategies under three typhoon duration schemes. **a** Actual typhoon tracks grouped in cluster #15. This is the best matched cluster while Typhoon Cora, the best matched track, has a track the most similar in shape to the track of Typhoon Fitow. **b** Predicted flood hydrographs using the first selection strategy. This strategy selects the flow characteristic curve of the best matched track. **c** Predicted flood hydrographs using the second selection strategy. This strategy selects the ensemble of all flow characteristic curves in the best matched cluster. We use three schemes to characterize typhoon duration. The first scheme spans between the arrival and departure time of a typhoon over the gridded zone. The second scheme spans between the start rising flow limb characteristics and the departure of the typhoon from the gridded zone. The third scheme spans between the start rising flow limb characteristics and the cessation of rainfall.

significantly improved over the last three decades, where errors have been reduced by two-thirds in just 25 years[42], and an averaged error <100 km has been achieved for the test typhoon events[43]. We expect that typhoon track forecasts for areas with topographic influence will be improved by the new prediction technology coupled with our CWB typhoon warnings before typhoon landfall. In this setting, input data of our model can be improved and more accurate flood hydrograph predictions can be made by our approach, which will provide new critical information for flood defense and water management.

We recognize that it is not possible to define the timing of rainfall cessation as well as the timing of typhoon departure from a watershed in advance. However, this information could be extracted simply from historical records and/or be estimated based on simple calculation, such as dividing the distance by the speed, with acceptable estimation accuracy. To improve the accuracy and reliability of flood hydrograph prediction, the next step will include the total rainfall amount and TC velocity. Although, in the present study, predictions are conducted two days ahead, extension of the prediction time

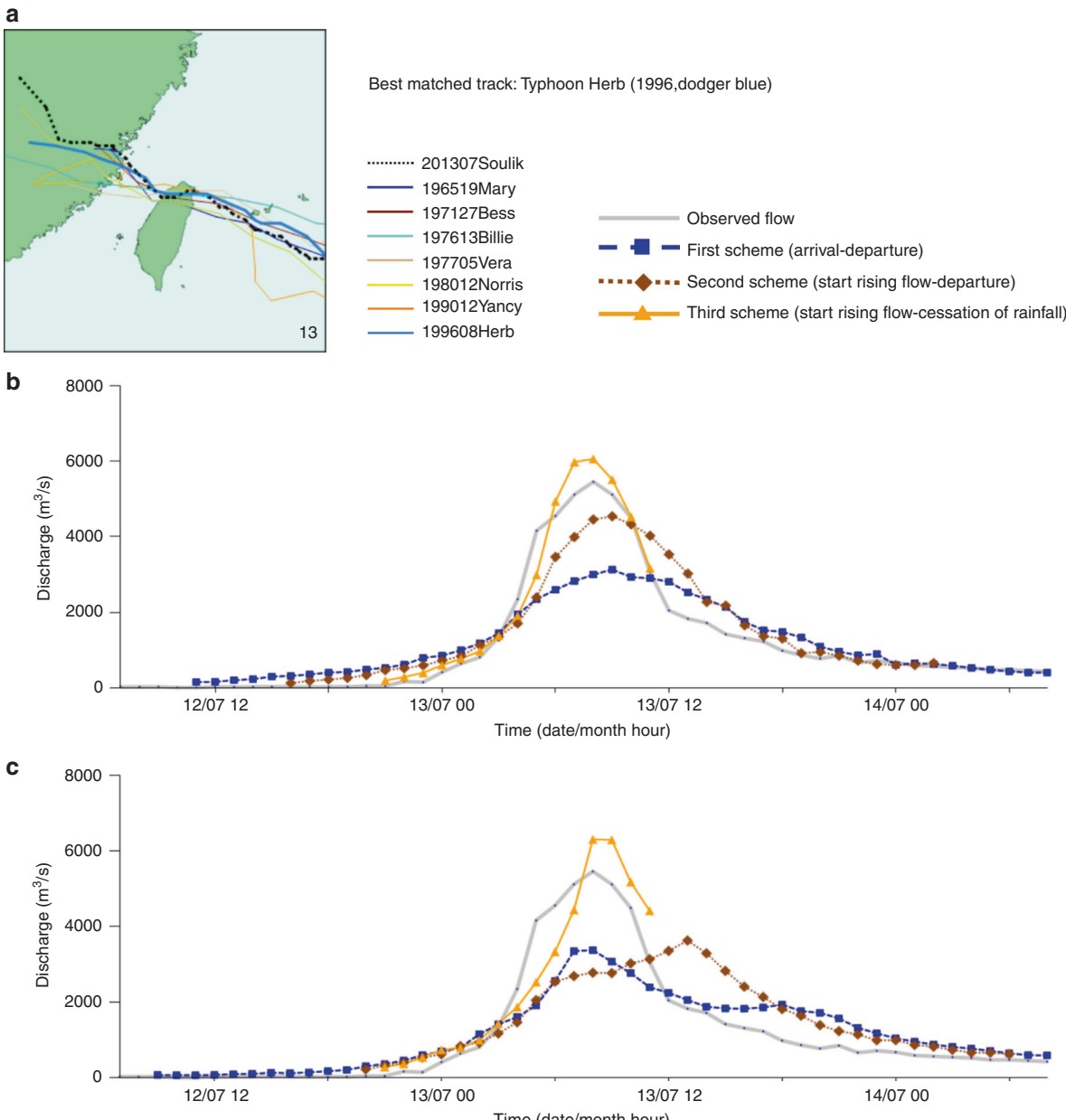

**Fig. 8 Predicted flood hydrographs for Typhoon Soulik.** Predictions were made using two flow characteristic curve selection strategies under three typhoon duration schemes. **a** Actual typhoon tracks grouped in cluster #13. This is the best matched cluster while Typhoon Herb is the best matched track. **b** Predicted flood hydrographs using the first selection strategy. **c** Predicted flood hydrographs using the second selection strategy.

interval is straightforward once improved data becomes available.

While typhoon tracks may change when crossing Taiwan, the proposed methodology allows for continual updates of flood hydrograph predictions based on recent track forecasts and total rainfall before and/or during typhoon passing over Taiwan. This approach generates real-time flood hydrograph predictions needed for reservoir flood control operation as well as water resources management. For example, flood peaks and timing demand for different reservoir operations. For example, a high flood peak and rapid filling of the reservoir may threaten dam safety. An unnecessary reduction in reservoir freeboard may threaten water supply requirements.

Uncertainty is intrinsic in the current state of hydro-metrological science. To capture uncertainty a more holistic and transdisciplinary science that represents important linkages between hydrology, metrology, and topography is needed in water resources management. Further, improved approaches and tools to communicate uncertainty and provide advanced warning for flood defense are more urgent than ever. Our study digitizes analog typhoon tracks for the first time. Now the vectorized track can be associated with hydrologic data and geographic characteristics to support flood hydrograph forecasting and flood defense. The framework we developed can serve as an intelligent diagnostic tool for hydro-meteorologists and can be used to characterize an approaching typhoon for making flood hydrograph predictions with real-time updates. This not only provides an adaptive early warning for flood defense as well as critical information for reservoir operation but also better communicates concerning uncertainty in support of decision and policy making to best serve our society. The methodology is simple to use and flexible with applications to tackling problems, ranging from

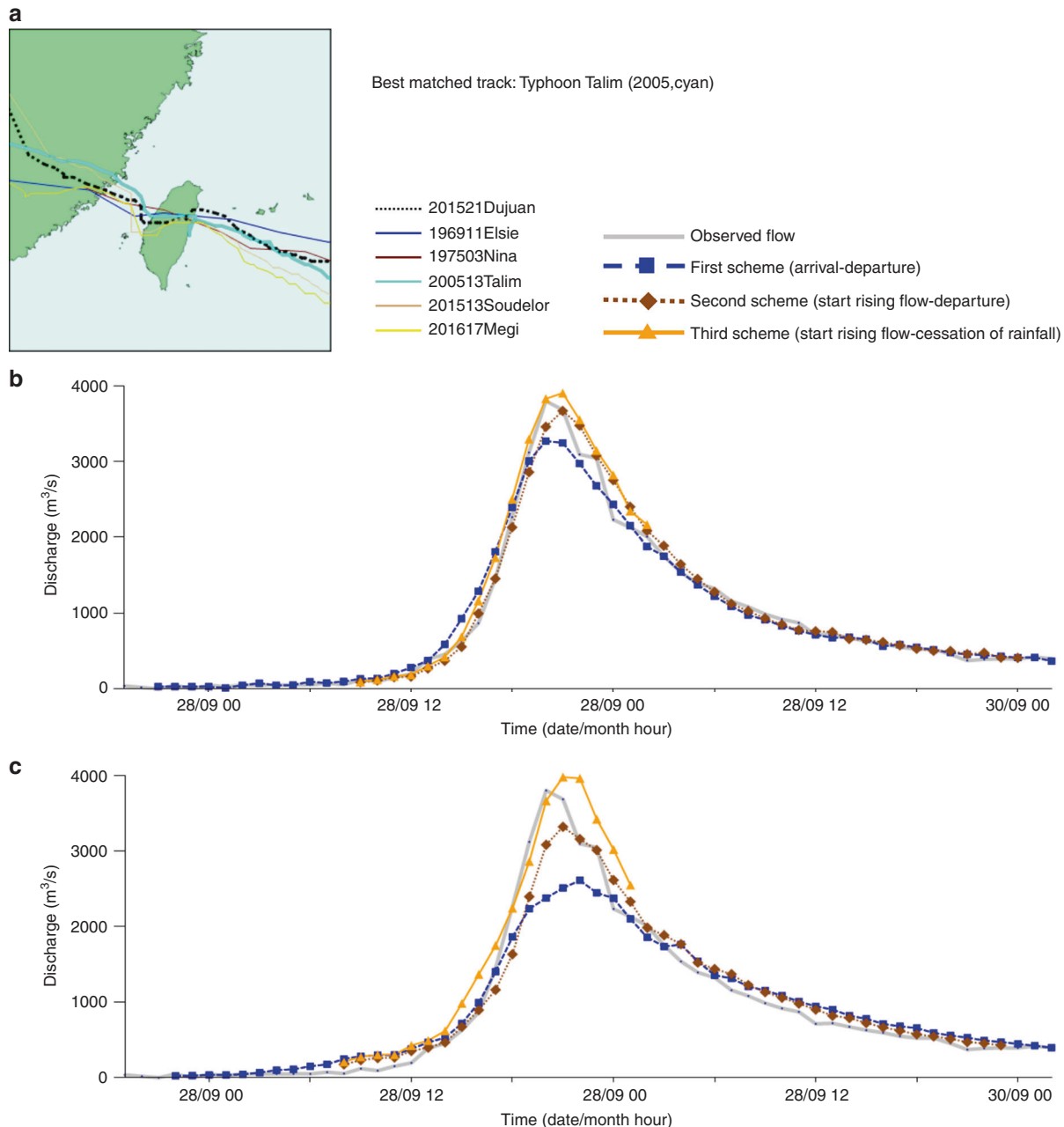

**Fig. 9 Predicted flood hydrographs for Typhoon Dujuan.** Predictions were made using two flow characteristic curve selection strategies under three typhoon duration schemes. **a** Actual typhoon tracks grouped in cluster #3. This is the best matched cluster while Typhoon Talim is the best matched track. **b** Predicted flood hydrographs using the first selection strategy. **c** Predicted flood hydrographs using the second selection strategy.

climatic change prediction extreme event series to financial markets.

## Methods

The proposed AI-based methodology has three stages: typhoon track vectorization and clustering; FCC extraction; and flood hydrograph prediction, shown as follows.

**Typhoon track vectorization and clustering**. A total of 97 typhoons were used to build the flood hydrograph prediction model, among them 87 typhoons were used for model training and the remaining 10 typhoons were used for model testing. In the first stage, typhoon tracks required an analog to digital conversion followed by clustering of track vectors. Individual typhoon tracks were projected onto a $5 \times 5$ geographic grid established between 116° and 126°E longitude and 20° and 30°N latitude. This grid included Taiwan and surrounding areas. To provide a watershed focus this study used the Shihmen Reservoir watershed where smaller grids were developed, i.e., the grid size was further reduced to four smaller ones including

huge ($2° * 2°$), large ($1° * 1°$), medium ($0.5° * 0.5°$), and small ($0.25° * 0.25°$) grids. Then a grid was marked if a typhoon passed through this grid. The size of a grid was determined mainly by the density of marked points present (the number of times a grid was marked in response to 87 typhoon events). The higher the density was, the smaller the size was. The largest grid size represented the lowest density while the smallest grid size represented the highest density, which allowed location specificity to connect typhoon tracks with watershed impacts produced by flooding. Accordingly, a total of 277 grids of different sizes constituted the gridded zone by resizing the 25 grids into finer grids, as shown in Fig. 1.

We next designed two rules to effectively vectorize typhoon tracks one by one over the gridded zone. For each typhoon, track passage over a grid element was identified. If a cluster had multiple tracks over the same grid element a positive weight value was assigned based on grid element size (1 for the biggest grid element, 2 for large grid elements, 3 for medium grid elements, and 4 for small grid elements); otherwise, grid elements were assigned a zero value (a zero-weight grid). Then, a weight diffusion process was carried out to address the issue of neighboring zero-weight grids that would be the most likely to reflect track variability. For large and medium weighted grid cells the adjacent zero-weight grids were re-assigned

values of 0.8 and 1.5, respectively. We also diffused the weight values over two successive layers adjacent to small weighted grids by assigning their zero-weight neighbors with 2.5 (within a radius of 1) and 1 (within a radius of 2, excluding grids weighted 2.5). As a result, each typhoon track could be converted into a digital vector of 277 variables (grids) with weight values (no shding–0, and gray to black shading with 0.8, 1, 1.5, 2, 2.5, and 3 weight values in Fig. 1d). We recognize that weight values are subjectively assigned based on trial and error; nevertheless, weights were used in optimization algorithms.

Cluster analysis provides a systematic way to categorize instances into subcategories. In this study, the AI-based SOM clustering method was implemented to classify the vectorized typhoon tracks. The SOM is a powerful tool to explore high dimensional data sets and is widely applied to clustering problems in various fields[44–48]. The SOM can extract patterns from large data sets with high dimensionality providing a method to form a two-dimensional topological map for visualizing and exploring data structures. The novelty here was to classify each vectorized typhoon track based not only on a few points of its trajectory as commonly used in previous studies[13,48] but also on its full track approaching close to the target watershed. The diffused vector of each typhoon track constituted model input, which would be mapped onto the two-dimensional array (neurons) of the SOM. Because the number of input variables is limited (87-diffused vectors) and the dimension of the data is large (277 grids), only three SOMs with small map sizes (i.e., $3 \times 3$, $4 \times 4$, $5 \times 5$) were trained to configure topological maps for selecting a suitable SOM topology for clustering purpose. The results of map size evaluation found that the $3 \times 3$ network could not distinctly present the classification while the $5 \times 5$ network resulted in a dispersion of few tracks in each cluster. In contrast, the SOM with $4 \times 4$ neurons provided the best mix of classification presentation and track numbers in clusters producing a typology supporting our analysis (Fig. 2). Consequently, we used a $4 \times 4$ SOM to analyze FCCs in this study. The benefit of this track clustering was the ability to match an approaching typhoon track with a typhoon track similar in shape present in the SOM expanding opportunities for prediction.

**FCC extraction**. The FCC of each typhoon event in the same cluster could be estimated based simply on the total flow volume converted directly from total rainfall. More precisely, an event-specific FCC could be developed by analyzing the overall flood hydrograph of the typhoon event using a normalization process. The normalization process used incorporated a dimensionless analysis that transformed the cumulative inflow volume and TD into a common scale ranging between 0 and 1. The curve of the normalized ratio of the cumulative inflow over time to the total inflow would start with a minimum value of 0 and continuously increased to a maximum value of 1 over time. Therefore, a FCC curve could then be tagged to the SOM cluster with which this typhoon was associated. The total rainfall and the duration of a typhoon event were two crucial factors significantly affecting the shape of its FCC. It was observed that TD might last for hours or days. We thus used three approaches to define effective TD with the goal of fitting the shape of event-specific FCCs using a comparison of shape.

The TD calculation was refined into three schemes. The first scheme focused on the arrival and departure time of each typhoon over the gridded zone. As a typhoon was entering the zone, it would commonly take one to three days to raise the flow, depending mainly on the speed (intensity) of the typhoon. Therefore, the rise in flow in the same cluster could be inconsistent at the early stage, which would make FCCs rather different in the same cluster. The second scheme emphasized the rising limb characteristics related to the departure of the typhoon from the gridded zone. As a typhoon was approaching the watershed and an obvious increase in flow was detected, the watershed was indeed influenced by the typhoon, where flow at this stage consistently increased to reach the peak and then dropped. The influence of typhoons on flow at this stage was complicated particularly where mountains were coupled with other complex geomorphological features. A particular issue is the change in intensity class from a typhoon category to a tropical depression, which was related to the localized influence of rainfall. The third scheme was centered on the rising limb characteristics related to the cessation of rainfall. After a typhoon passed through the watershed, rainfall might cease or keep falling subject to external circulation. We noticed that the FCCs in each neuron would become more consistent in shape if the duration between the rise of flow and the cessation of rainfall was adopted. This provided good evidence that flow patterns in the same cluster (neuron) were closely related to the corresponding typhoon tracks and similar typhoon tracks would produce similar FCCs.

We recognized that it was not possible to define the timing of rainfall cessation in advance. We further noticed that the timing for flow increase related to the timing of typhoon departure from a watershed could be extracted simply from historical records and/or be estimated based on simple calculation, such as dividing the distance by the speed, with acceptable estimation accuracy.

The TD calculation method we developed effectively addressed concentration time in a flood hydrograph, particularly considering the inclusion of rising limb characteristics.

**Flood hydrograph prediction**. In this study, flood hydrograph prediction offered by our proposed methodology depends mainly on four components including the forecasted typhoon track, forecasted total rainfall, typhoon track classification, and

FCC extraction. For an approaching typhoon, the representative FCC of a cluster can be determined by visually identifying a matched cluster that had a typhoon track the most similar to the forecasted typhoon track announced by the CWB. Consequently, the flood hydrograph of the approaching typhoon can be derived based on the representative FCC of the best matched cluster together with the forecasted total rainfall.

## Data availability
The full data that support the findings of this study are available at http://www.icwe.tku. edu.tw/TyphoonData/.

## Code availability
The custom codes and algorithms developed/used in this study are available from Prof. Li-Chiu Chang (changlc@mail.tku.edu.tw) upon reasonable request.

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

## Acknowledgements

This research was partly supported by the North Region Water Resources Office, Water Resources Agency (WRA), Ministry of Economic Affairs, Taiwan (MOEA-WRA1030184). L.C.C. and F.J.C. were also supported by the Ministry of Science and Technology, Taiwan (NSC 101-2313-B-032-002-MY3 and 103-2313-B-002-016-MY3, respectively). We thank the WRA and the Central Weather Bureau (CWB) for providing access to their data. The authors would like to thank the Editors and anonymous Reviewers for their constructive comments that are greatly contributive to improving the manuscript.

## Author contributions

L.C.C. and F.J.C. designed the study and performed analyses. F.H.T. and S.N.Y. assisted in data analysis. L.C.C., F.J.C., and E.E.H. contributed to the interpretation of the results and writing the manuscript with discussion and feedback from T.H.C. and S.N.Y.

## Competing interests

The authors declare no competing interests.
