## [Peer Review File · Nature Communications]

Reviewers' comments:

Reviewer #1 (Remarks to the Author):

1. The manuscript presents excavating typhoon tracks to AI-based flood early warnings, which is interesting. The subject addressed is within the scope of the journal.
2. However, the manuscript, in its present form, contains several weaknesses. Appropriate revisions to the following points should be undertaken in order to justify recommendation for publication.
3. Full names should be shown for all abbreviations in their first occurrence in texts. For example, AI in p.1, etc.
4. For readers to quickly catch your contribution, it would be better to highlight major difficulties and challenges, and your original achievements to overcome them, in a clearer way in abstract and introduction.
5. It is shown in the reference list that the authors have several publications in this field. This raises some concerns regarding the potential overlap with their previous works. The authors should explicitly state the novel contribution of this work, the similarities and the differences of this work with their previous publications.
6. It is mentioned in p.1 that an AI-based intelligent modeling approach is adopted for predictions of typhoon path and the potential for flooding. What are the other feasible alternatives? What are the advantages of adopting this particular approach over others in this case? How will this affect the results? More details should be furnished.
7. It is mentioned in p.2 that the Shihmen Reservoir watershed is adopted as the case study. What are other feasible alternatives? What are the advantages of adopting this particular case study over others in this case? How will this affect the results? The authors should provide more details on this.
8. It is mentioned in p.5 that historical records of 1971 to 2015 are taken. Why are more recent data not included in the study? Is there any difficulty in obtaining more recent data? Are there any changes to situation in recent years? What are its effects on the result?
9. It is mentioned in p.5 that total rainfall, typhoon track, the date and time warnings issued, hourly reservoir inflow, and outflow peak data are adopted as the data set for the prediction model. What are other feasible alternatives? What are the advantages of adopting these parameters over others in this case? How will this affect the results? The authors should provide more details on this.
10. It is mentioned in p.6 that a track vectorization method is adopted to produce digitized typhoon tracks. What are the other feasible alternatives? What are the advantages of adopting this particular method over others in this case? How will this affect the results? More details should be furnished.
11. It is mentioned in p7 that self-organized map is adopted to cluster the 87 tracks selected for model training into a map. What are the other feasible alternatives? What are the advantages of adopting this particular soft computing technique over others in this case? How will this affect the results? More details should be furnished.

12. It is mentioned in p.10 that flow characteristic curve is adopted to provide slopes for the rising and recession limbs of the hydrograph. What are the other feasible alternatives? What are the advantages of adopting this particular approach over others in this case? How will this affect the results? More details should be furnished.

13. It is mentioned in p.11 that three schemes are adopted to classify flow characteristic curves. What are other feasible alternatives? What are the advantages of adopting these schemes over others in this case? How will this affect the results? The authors should provide more details on this.

14. It is mentioned in p.14 that two strategies are adopted for flow characteristic curve prediction. What are other feasible alternatives? What are the advantages of adopting these strategies over others in this case? How will this affect the results? The authors should provide more details on this.

15. Some key parameters are not mentioned. The rationale on the choice of the particular set of parameters should be explained with more details. Have the authors experimented with other sets of values? What are the sensitivities of these parameters on the results?

16. Some assumptions are stated in various sections. Justifications should be provided on these assumptions. Evaluation on how they will affect the results should be made.

17. The discussion section in the present form is relatively weak and should be strengthened with more details and justifications.

18. Moreover, the manuscript could be substantially improved by relying and citing more on recent literatures about contemporary real-life case studies of soft computing techniques in hydrologic prediction such as the followings:

☐ Yaseen, Z.M., et al., "An enhanced extreme learning machine model for river flow forecasting: state-of-the-art, practical applications in water resource engineering area and future research direction," *Journal of Hydrology* 569: 387-408 2019.

☐ Cheng, C.T., et al., "Three-person multi-objective conflict decision in reservoir flood control," *European Journal of Operational Research* 142 (3): 625-631 2002.

☐ Moazenzadeh, R., et al., "Coupling a firefly algorithm with support vector regression to predict evaporation in northern Iran," *Engineering Applications of Computational Fluid Mechanics* 12 (1): 584-597 2018.

☐ Wu, C.L., et al., "Rainfall-Runoff Modeling Using Artificial Neural Network Coupled with Singular Spectrum Analysis", *Journal of Hydrology* 399 (3-4): 394-409 2011.

☐ Ghorbani, M.A., et al., "Forecasting pan evaporation with an integrated Artificial Neural Network Quantum-behaved Particle Swarm Optimization model: a case study in Talesh, Northern Iran," *Engineering Applications of Computational Fluid Mechanics* 12 (1): 724-737 2018.

☐ Chau, K.W., et al., "Use of Meta-Heuristic Techniques in Rainfall-Runoff Modelling" *Water* 9(3): article no. 186, 6p 2017.

19. In the conclusion section, the limitations of this study, suggested improvements of this work and future directions should be highlighted.

Reviewer #2 (Remarks to the Author):

Review of Chang et al.

General comments: this is a very interesting study that explores the use of AI/SOM clustering for typhoon-induced flooding forecasting. However, there are several unanswered critical issues and overstatements that need to be addressed before acceptance can be recommended:

1. It is unclear how the authors can claim to have improved predictions of the typhoon tracks. Categorization of the CWB forecast track into one of the clusters does not by itself make the track forecast better. The 3 forecast examples happened to be the cases that the CWB forecast tracks are reasonably predicted. What if the CWB track is incorrectly forecasted to a significantly different neurons? Can the authors show how bad if such a bad forecast will lead to flood forecasting?
2. More generally, given the inherent uncertainties in typhoon track forecasting, and the critical dependence of flooding on typhoon track and duration forecasting, it is unclear how the proposed method will incorporate the forecast uncertainties as now most operational weather prediction centers are moving from deterministic (single-track) forecasting to probabilistic (ensemble) forecasting? One example of such diverse track uncertainty and the consequence to extreme flooding forecasting can be seen for Typhoon Morakot (2009), see reference such as Zhang et al. (2010, *Weather and Forecasting*, page 1816-1825).
3. The usage of the proposed method appears to be rather limited to the 2-day range of forecast lead time as the past analog tracks shown in Fig.2a. How can the method be extended to longer lead times since in many remote areas or reservoir planning a 2-day lead time warning will not be sufficient for evacuation and flood control? With a potential big domain than Figure 2a needs to be considered, would this make the SOM approach less concentrated since there are a lot more potential track scenarios with longer lead times?
4. It is unclear how the authors can claim the new method improves over the existing warning system by a longer lead time for focused flood defense. What is the existing system, and how to systematically validate your claims?
5. More philosophically, how can be the proposed technique to be generalizable for other applications, in other areas, and with the inclusion of probabilities that are crucial to any extreme events forecasting?

Reviewer #3 (Remarks to the Author):

This paper uses a method of analogs to predict typhoon tracks and flooding properties. In particular, they preprocess the historical track data using self-organized maps, a method from the field of artificial intelligence (AI). The idea of applying algorithms from the field of AI to these important problems of typhoon and flood prediction is very worthy and promising.

The paper is technically solid, indicating careful application of these methods and including detailed explanation of each step, including how they digitally represented the typhoon track data.

This reviewer will abstain from commenting on the quality of the results in the field of flood prediction, not being an expert in the area. It is difficult to draw conclusions without an empirical comparison to other approaches to this task (including a simple baseline) from the relevant typhoon and flood literature.

From the AI perspective, while the AI tools seem to have been properly applied, the choice of AI tools would need to be justified via empirical comparisons to other methods used in the literature. For example, extreme storm track prediction has recently been done by a variety of supervised machine learning techniques, such as convolutional neural networks, using both storm track data and SST reanalysis data (wind, pressure) around the storm.

Reviewer #1 (Remarks to the Author):

1. *The manuscript presents excavating typhoon tracks to AI-based flood early warnings, which is interesting. The subject addressed is within the scope of the journal.*

Response: Thank you for taking time and effort to review our study and providing many valuable comments suggestions.

2. *However, the manuscript, in its present form, contains several weaknesses. Appropriate revisions to the following points should be undertaken in order to justify recommendation for publication.*

Response: We have revised our manuscript accordingly.

3. *Full names should be shown for all abbreviations in their first occurrence in texts. For example, AI in p.1, etc.*

Response: Done.

4. *For readers to quickly catch your contribution, it would be better to highlight major difficulties and challenges, and your original achievements to overcome them, in a clearer way in abstract and introduction.*

Response: Thank you for the constructive suggestion. This AI-based approach is a new intelligent diagnostic tool for hydro-meteorologists to improve flood early warnings through providing flood hydrograph predictions, which could not be appropriately achieved by traditional rainfall-runoff predictions. As suggested, the following statements have been incorporated into the revised manuscript for highlighting the challenges and enhancing our achievements.

In Abstract:

This revolutionizes traditional rainfall-runoff approaches and supports site-specific typhoon track–flood hydrograph prediction. With this site specificity, it is now possible to predict flood hydrographs before typhoon landfall supporting early warnings for reservoir management and flood defense. (Lines 31-34)

In Main:

The high variability in typhoon tracks over Taiwan and the complex rainfall patterns produced by the island's topography make accurate flood forecasting challenging using existing models^{9,10}. Model advancements are requested to provide the lead time needed to implement flood defense procedures and adjust reservoir capacity for flood control that meet other water requirements for domestic, industrial, agricultural and hydropower generation purposes. A recent study of rainfall-runoff modeling based on remote rainfall information indicated that reliable real-time flood forecasts could be

obtained only up to six hours ahead during typhoon event³. Therefore, it is imminent to revolute traditional rainfall-runoff approaches and develop site-specific typhoon track–flood hydrograph prediction several days before typhoon landfall. (Lines 72-81)

5. It is shown in the reference list that the authors have several publications in this field. This raises some concerns regarding the potential overlap with their previous works. The authors should explicitly state the novel contribution of this work, the similarities and the differences of this work with their previous publications.

Response: OK! We have explicitly stated the novel contribution of this work in the revised manuscript. Indeed, we do have a number of studies that develop and /or use various AI techniques (e.g. machine learning, ANNs) to solve various hydrological issues and the results demonstrate the applicability and reliability of AI techniques for various complex hydrological issues, especially short-term (hourly) flood forecasts. We would like to note from our recent study on rainfall-runoff modeling that reliable flood forecasts could be obtained only a few hours ahead (Chang & Tsai, 2016).

Reference:

Chang, F. J., & Tsai, M. J. A nonlinear spatio-temporal lumping of radar rainfall for modeling multi-step-ahead inflow forecasts by data-driven techniques. *J. Hydrol.* 535, 256-269 (2016).

6. It is mentioned in p.1 that an AI-based intelligent modeling approach is adopted for predictions of typhoon path and the potential for flooding. What are the other feasible alternatives? What are the advantages of adopting this particular approach over others in this case? How will this affect the results? More details should be furnished.

Response: This study is the first attempt to make two-day-ahead-flood hydrograph predictions based on typhoon tracks through the proposed AI-based approach. To our best knowledge, there is no other feasible alternatives that could make two-day-ahead flood hydrograph predictions without given rainfall patterns. One could just use the two-day-ahead (48 hours) forecast of rainfall pattern as an input to make two-day-ahead-flood hydrograph predictions while the great uncertainty of the forecasted rainfall pattern would destroy the reliability of flood hydrograph predictions.

Advantages of the proposed method include:

1. With regard to current technology, hydrograph prediction based on typhoon track is more accurate than based on the hourly rain pattern.
2. Typhoon track prediction could be obtained one or two days before landfall. Thus we could make an early flood warning several days before landfall.
3. The proposed method allows small errors in typhoon track prediction (the

predicted typhoon track might be classified into the same cluster if the error is not too large, and therefore the forecast results of the flood hydrograph are similar). Thus, the proposed method is a robust error- tolerant method.

To assess the reliability and accuracy of the AI-based approach, we compared the prediction results with those of a commonly used conceptual rainfall-runoff model, i.e. the storage function model (SFM)^{26,27} (Lines 283-285)

7. It is mentioned in p.2 that the Shihmen Reservoir watershed is adopted as the case study. What are other feasible alternatives? What are the advantages of adopting this particular case study over others in this case? How will this affect the results? The authors should provide more details on this.

Response: The Shihmen Reservoir is a pivotal multi-objective reservoir that provides flood protection and a water supply of more than 800 million m³/year to meet the domestic, agricultural, and industrial needs of the Taipei metropolitan area, thus requiring careful management. (Lines 51-54)

The proposed methodology could be easily implemented in other watersheds in Taiwan as well as most of the Asian areas suffering from typhoon attack. All the necessary information (data set, methods as well as related parameters) is clearly presented in the revised manuscript or the supplementary file.

8. It is mentioned in p.5 that historical records of 1971 to 2015 are taken. Why are more recent data not included in the study? Is there any difficulty in obtaining more recent data? Are there any changes to situation in recent years? What are its effects on the result?

Response: OK, we have added and analyzed the results of four typhoon events that had happened in the last three years. We have provided a number of statements and analytical results in the supplementary file to clarify the concerns. Thanks!

9. It is mentioned in p.5 that total rainfall, typhoon track, the date and time warnings issued, hourly reservoir inflow, and outflow peak data are adopted as the data set for the prediction model. What are other feasible alternatives? What are the advantages of adopting these parameters over others in this case? How will this affect the results? The authors should provide more details on this.

Response: As said, the total rainfall, typhoon track, the issuance date and time of each warning constitute model input information, and data of hourly reservoir inflow and outflow peak are used to validate the goodness-of-fit of the proposed methods. The advantage of adopting these parameters is mainly because that they could be on-line obtained from government agencies, which make the system operate real-time and the

goodness-of-fit could be on-line obtained and updated.

10. *It is mentioned in p.6 that a track vectorization method is adopted to produce digitized typhoon tracks. What are the other feasible alternatives? What are the advantages of adopting this particular method over others in this case? How will this affect the results? More details should be furnished.*

Response: To our best knowledge, this is the first time to vectorize typhoon tracks for use in predicting flood hydrograph. This track vectorization method allowed a decomposition of a continuous (analog) track into a discontinuous grid vector without losing key features (e.g. direction, speed and duration). We produced digitized typhoon tracks for the 97 events using the vectorization procedure that assigned the values in 277 grids for each investigative typhoon track.

11. *It is mentioned in p7 that self-organized map is adopted to cluster the 87 tracks selected for model training into a map. What are the other feasible alternatives? What are the advantages of adopting this particular soft computing technique over others in this case? How will this affect the results? More details should be furnished.*

Response: Techniques for typhoon track classification are presently available, for example using K-means and fuzzy clustering as prediction tools. But these methods must be improved to refine track vectors and support location-specific analysis. Refinement steps include analog to digital conversion of typhoon movement into digital track vectors, characterization of track variability, and analysis of track interactions with terrain. The SOM was used to cluster the 87 tracks selected for model training into a map that displayed the 87 clustered tracks in a topological map. The clustering results grouped similar tracks in the same cluster (neuron). It was noted that the tracks behaved more consistently between adjacent neurons than non-adjacent neurons.

To adopt the valuable comments, the following statements have been added.

As known, the SOM is a powerful tool to form a two-dimensional topological map where similar tracks would be clustered in the same neuron and relative tracks would be clustered in the adjacent neurons. The benefit of this track clustering was the ability to match an approaching typhoon track with a typhoon track similar in shape present in the SOM. (Lines 313-317)

12. *It is mentioned in p.10 that flow characteristic curve is adopted to provide slopes for the rising and recession limbs of the hydrograph. What are the other feasible alternatives? What are the advantages of adopting this particular approach over others in this case? How will this affect the results? More details should be furnished.*

Response: The advantage of this approach is that the calculation of flow hydrographs would no longer require observed rainfall–runoff data that is only available post-event. Further, using this approach a flood hydrograph could be converted from a FCC cluster and an expected typhoon track supporting the development of advanced flood warnings.

13. It is mentioned in p.11 that three schemes are adopted to classify flow characteristic curves. What are other feasible alternatives? What are the advantages of adopting these schemes over others in this case? How will this affect the results? The authors should provide more details on this.

Response: Thank you for the insightful comment. Yes, considering this time dependence, we classified FCCs by three schemes based on TD data that identified: a) the arrival and departure time of a typhoon over the gridded zone; b) the start rising flow limb characteristics and the departure of the typhoon from the gridded zone; and c) the start rising limb characteristics and the cessation of rainfall. This approach allowed us to explore how typhoon stage/duration would influence the group behavior of FCCs in each neuron. The FCCs in each cluster were then more similar in shape when effective TD was reduced. For example, the difference between start-end hours in neuron #5 was reduced from 48-167 (Fig. 3a) to 35-120 (Fig. 3b), and to 23-66 (Fig. 3c), respectively. This supported the notion that a close relationship between typhoon tracks and their corresponding flood hydrographs could be obtained in each neuron when the TD calculation scheme was implemented.

14. It is mentioned in p.14 that two strategies are adopted for flow characteristic curve prediction. What are other feasible alternatives? What are the advantages of adopting these strategies over others in this case? How will this affect the results? The authors should provide more details on this.

Response: Thank you for the valuable comment.

We develop a novel AI-based (Artificial Intelligence, AI) modeling approach for making site-specific typhoon track–flood hydrograph prediction before typhoon landfall. The new intelligent modeling approach presented here provides hydro-meteorologists and watershed managers with a warning several days ahead of landfall as well as provides real-time updates for flood hydrograph prediction during typhoon events. The approach improves the existing Taiwan flood warning system by providing a longer lead time for focused flood defense. To our best knowledge, there is no feasible alternatives that could make reliable long-horizontal (more than 6 hours in our case) flood prediction. We have added the following statement to enhance the advantage of our approach.

A recent study of rainfall-runoff modeling based on remote rainfall information indicated that reliable real-time flood forecasts could be obtained only up to six hours ahead during typhoon event³. (Lines 77-79)

15. *Some key parameters are not mentioned. The rationale on the choice of the particular set of parameters should be explained with more details. Have the authors experimented with other sets of values? What are the sensitivities of these parameters on the results?*

Response: Thanks. We developed a new intelligent diagnostic tool for hydro-meteorologists to improve early warnings through providing flood hydrograph predictions. This study is mainly a data-driven modeling approach, where the parameters have been calibrated and validated based on the monitoring datasets. The key parameters (such as the map size of the SOM, the number of grids and the weighted grid values) are given in the Method Section.

16. *Some assumptions are stated in various sections. Justifications should be provided on these assumptions. Evaluation on how they will affect the results should be made.*

Response: We have tried our best to provide the justification and related assumptions in various sections. Also, we have added some supplementary materials to better justify the goodness-of-fit of our proposed AI-based approach.

17. *The discussion section in the present form is relatively weak and should be strengthened with more details and justifications.*

Response: OK! We have added more discussions (marked in blue) in the revised manuscript to strengthen the methodology, the applicability and accuracy of the proposed methods, as well as the uncertainty issue.

18. *Moreover, the manuscript could be substantially improved by relying and citing more on recent literature about contemporary real-life case studies of soft computing techniques in hydrologic prediction such as the followings:*

Yaseen, Z.M., et al., “An enhanced extreme learning machine model for river flow forecasting: state-of-the-art, practical applications in water resource engineering area and future research direction,” *Journal of Hydrology* 569: 387-408 2019.

Cheng, C.T., et al., “Three-person multi-objective conflict decision in reservoir flood control,” *European Journal of Operational Research* 142 (3): 625-631 2002.

Moazenzadeh, R., et al., “Coupling a firefly algorithm with support vector regression to predict evaporation in northern Iran,” *Engineering Applications of Computational Fluid Mechanics* 12 (1): 584-597 2018.

Wu, C.L., et al., “Rainfall-Runoff Modeling Using Artificial Neural Network Coupled with Singular Spectrum Analysis”, *Journal of Hydrology* 399 (3-4): 394-409 2011.

Ghorbani, M.A., et al., “Forecasting pan evaporation with an integrated Artificial Neural Network Quantum-behaved Particle Swarm Optimization model: a case study in Talesh, Northern Iran,” *Engineering Applications of Computational Fluid Mechanics* 12 (1): 724-737 2018.

Chau, K.W., et al., “Use of Meta-Heuristic Techniques in Rainfall-Runoff Modelling” *Water* 9(3): article no. 186, 6p 2017.

Response: Thanks! We have added the most relevant literatures as suggested. Some of recent literatures have also been added.

19. *In the conclusion section, the limitations of this study, suggested improvements of this work and future directions should be highlighted.*

Response: OK. We have added more statements regarding the limitation and future directions in the Discussion section.

Reviewer #2 (Remarks to the Author):

Review of Chang et al.

General comments: this is a very interesting study that explores the use of AI/SOM clustering for typhoon-induced flooding forecasting. However, there are several unanswered critical issues and overstatements that need to be addressed before acceptance can be recommended:

Response: Thank you for your interest in our study and for providing many insightful comments and constructive suggestions.

1. *It is unclear how the authors can claim to have improved predictions of the typhoon tracks. Categorization of the CWB forecast track into one of the clusters does not by itself make the track forecast better. The 3 forecast examples happened to be the cases that the CWB forecast tracks are reasonably predicted. What if the CWB track is incorrectly forecasted to a significantly different neuron? Can the authors show how bad if such a bad forecast will lead to flood forecasting?*

Response: We agree that “categorization of the typhoon tracks does not make the track forecast better”, and therefore delete the statement. We have added a paragraph to discuss how bad flood forecasting would be if such a bad forecast is made. We

demonstrated that the constructed SOM could be more valuable, as compared with other clustering methods!

Traditional rainfall-runoff models only could predict the flood time series gradually (step-by-step) based on given (known) rainfall patterns. Unlike traditional approaches, this study intends to prove that the flood hydrograph is highly correlated with the typhoon track, and we can make site-specific typhoon track-flood hydrograph prediction several days before typhoon landfall.

It is not uncommon to incorrectly predict typhoon track, consequently the track would be projected onto different neurons and result in poor flood forecasting. As known, the SOM is a powerful tool to form a two-dimensional topological map where similar tracks would be clustered in the same neuron and relative tracks would be clustered in the adjacent neurons. The benefit of this track clustering was the ability to match an approaching typhoon track with a typhoon track similar in shape present in the SOM. Nevertheless, in case of incorrect track forecast (but within certain range), it would be clustered into neighboring neurons and produced relatively good predictions. Thus, the proposed method could tolerate small errors in typhoon track prediction, and it is a robust error-tolerant approach. We present a recent case, Typhoon Lekima in 2019, which had a poorly predicted track, with an error of 80 km away from the north of Taiwan. The original predicted track should be mapped into #15 neuron, while the actual track is fallen into the #16 neuron. The projected results of original predicted track and actual track are given in Supplementary Figure 2. We notice that inaccurate track prediction does cause some differences in both the forecasted flood peak and the occurrence timing of a peak while these differences fall within a small (acceptable) range. Thus, the SOM clustering method deals with false predictions in an error-tolerant manner. (Lines 312-329)

2. More generally, given the inherent uncertainties in typhoon track forecasting, and the critical dependence of flooding on typhoon track and duration forecasting, it is unclear how the proposed method will incorporate the forecast uncertainties as now most operational weather prediction centers are moving from deterministic (single-track) forecasting to probabilistic (ensemble) forecasting? One example of such diverse track uncertainty and the consequence to extreme flooding forecasting can be seen for Typhoon Morakot (2009), see reference such as Zhang et al. (2010, Weather and Forecasting, page 1816-1825).

Response: Thank you very much for providing this insightful and constructive comment. We note that typhoon track forecasting is not our focus, while the uncertainties of typhoon track forecasting do affect the usefulness of our approach. To consider and incorporate forecast uncertainties, we used the SOM to cluster typhoon

tracks. The clustering results grouped similar tracks in the same cluster (neuron) and formed a topological map that the tracks behaved more consistently between adjacent neurons than non-adjacent neurons.

Your valuable and insightful comment, “as now most operational weather prediction centers are moving from single-track forecasting to ensemble forecasting”, inspires our thought and instruction. Our approach could adopt the results of ensemble forecasting. As matter of fact, we have done this. We have two strategies to make the FCC (flood pattern) prediction: the best matched typhoon track and the average of all FCCs (or, two or more FCCs) in this matched neuron. The best matched typhoon track could be regarded as the single-track forecasting while the average of all FCCs in this matched neuron could be regarded as the ensemble forecasting. Using either approach the prediction of the flood hydrograph of an approaching typhoon could be made. We have incorporated this important finding and re-polished our manuscript.

3. The usage of the proposed method appears to be rather limited to the 2-day range of forecast lead time as the past analog tracks shown in Fig.2a. How can the method be extended to longer lead times since in many remote areas or reservoir planning a 2-day lead time warning will not be sufficient for evacuation and flood control? With a potential big domain than Figure 2a needs to be considered, would this make the SOM approach less concentrated since there are a lot more potential track scenarios with longer lead times?

Response: Thank you for the insightful and valuable comments.

We would like to note that a forecast lead time of two days is set based mainly on the reliability of current typhoon prediction technologies as well as the catchment characteristic of the Shihmen Reservoir in this case study. The following statements have been added.

For the Shihmen Reservoir watershed, the impact of a typhoon on the watershed would last for 28 hours on average and the average time interval between the arrival and departure of a typhoon over the gridded zone would be 56 hours (based on 87 typhoon events). For flood control, if the water discharges from 245 m (full reservoir storage) 240 m, it would take about 11 hours at a rate of 1,000 cms and about 37 hours at a rate of 300 cms. Therefore, the prediction of flood hydrograph with a lead time of two days prior to typhoon landfall would be satisfactory for flood control as well as water management. (Lines 179-186)

We agree that a longer lead time would be required for large watersheds. The proposed method can be extended through enlarging the designed domain (our original domain was projected on a 5 x 5 geographic grid spanning between 116°-126°E longitude and 20°-30°N latitude). However, this proposed method would be

more suitable for island countries such as Taiwan, Japan, and the Philippines, whose catchment areas are small (less than a couple of thousand km²) and flash floods commonly occur within one day. For a large catchment area, its flow from upstream to downstream might take several days such that the flood problem may more suitably be solved by other alternative methods. For instance, for the Three Gorges Reservoir in China, its rainfall and runoff variables are non-stationary and thus it is difficult to model the notoriously non-linear rainfall-runoff pattern. One of our recent study indicated the travel time of flow running from each upstream station to the Three Gorges Reservoir varied between 6h and 48h (Zhou et al., 2019). Thus, the forecasts could not be properly derived based solely on rainfall-runoff patterns, where the upstream condition and human activities could be much more critical than the local rainfall information. Indeed, accurate long-horizon flood forecasting is still very challenging, which requires much more intensive and innovative studies.

Reference:

Zhou, Y., Guo, S., & Chang, F. J. (2019). Explore an evolutionary recurrent ANFIS for modelling multi-step-ahead flood forecasts. *Journal of hydrology*, 570, 343-355.

4. It is unclear how the authors can claim the new method improves over the existing warning system by a longer lead time for focused flood defense. What is the existing system, and how to systematically validate your claims?

Response: Thank for the insightful comments. For the existing forecast system, we do have a number of studies regarding the flood defense of the Shihmen Reservoir while all the studies we have had could only provide predictions few hours ahead (less than 6 hours in most cases) based on the rainfall-runoff modelling approach. As known, the rainfall-runoff process would only take less than a couple of hours. This is mainly because a reliable prediction of rainfall pattern at a longer lead time could not be obtained.

Advantages of the proposed method include:

1. With regard to current technology, hydrograph prediction based on typhoon track is more accurate than based on the hourly rain pattern.
2. Typhoon track prediction could be obtained one or two days before landfall. Thus we could make an early flood warning several days before landfall.
3. The proposed method allows small errors in typhoon track prediction (the predicted typhoon track might be classified into the same cluster if the error is not too large, and therefore the forecast results of the flood hydrograph are similar). Thus, the proposed method is a robust error- tolerant method.

Disadvantages of traditional method include:

1. The model commonly requires current rainfall and/or runoff as inputs, and

consequently it could only provide short-term (a couple of hours) forecasts.

2. Most of the conceptual models commonly have large prediction errors of flood peak due to their deterministic parameters.

To adopt your valuable suggestion, we have added simulation results based on a commonly used conventional rainfall-runoff model to assess the goodness-of-fit of our proposed method in making flood hydrograph prediction. We found that the predicted flood hydrographs of the 10 test events generally matched the actual flood hydrographs providing a long lead time (e.g. several days), with acceptable variation in the timing and volume of peak flows. This is a major improvement of existing prediction models (e.g. physical, conceptual, and data-driven) that focus on rainfall-runoff mechanisms providing a short lead time (e.g. one- to six-hours).

5. More philosophically, how can be the proposed technique to be generalizable for other applications, in other areas, and with the inclusion of probabilities that are crucial to any extreme events forecasting?

Response: Thank you very much for providing the constructive comments. Basically, the paper develops an innovative model that produces a new principle area where the practice can be applied to Taiwan. Principles are generalized, and practices are site specific. The methodology is simple to use and flexible with applications to tackling problems, ranging from climatic change prediction extreme event series to financial markets.

We have added a few statements to enhance (promote) its applicability and generalizability.

In abstract:

The analysis of typhoon tracks in Taiwan provides a useful microcosm for broader application to Pacific typhoons. (Lines 23-24)

This revolutionizes traditional rainfall-runoff approaches and supports site-specific typhoon track–flood hydrograph prediction. With this site specificity, it is now possible to predict flood hydrographs before typhoon landfall supporting early warnings for reservoir management and flood defense. (Lines 31-34)

Reviewer #3 (Remarks to the Author):

This paper uses a method of analogs to predict typhoon tracks and flooding properties. In particular, they preprocess the historical track data using self-organized maps, a method from the field of artificial intelligence (AI). The idea of applying

algorithms from the field of AI to these important problems of typhoon and flood prediction is very worthy and promising.

Response: We sincerely thank you for recognizing our study and providing insightful comments and constructive suggestions. Your recognition highly encourages us. All your concerns and brilliant suggestions have been fully addressed and/or incorporated into the revised manuscript.

The paper is technically solid, indicating careful application of these methods and including detailed explanation of each step, including how they digitally represented the typhoon track data.

Response: Thanks!

This reviewer will abstain from commenting on the quality of the results in the field of flood prediction, not being an expert in the area. It is difficult to draw conclusions without an empirical comparison to other approaches to this task (including a simple baseline) from the relevant typhoon and flood literature.

Response: Thank you for your valuable comment. Our proposed AI-based method is a new and revolutionary approach, which only requires the typhoon track and the corresponding rainfall amount to make flood hydrographs several days before typhoon landfall. To adopt your valuable suggestion, we have implemented a commonly used method as a comparative model, i.e. the storage function model (SFM), in this revised manuscript. We notice that this traditional rainfall-runoff approach requires a rainfall pattern to produce the flood hydrograph, and thus it is more like a simulation method, rather than a predictor. Thus, its simulation results could only be used to learn the goodness-of-fit of our proposed method in making-flood hydrograph prediction. The analytical results are added and shown as follows. To assess the reliability and accuracy of the AI-based approach, we compared the prediction results with those of a commonly used conceptual rainfall-runoff model, i.e. the storage function model (SFM)^{26,27} (Lines 283-285)

As known, the SFM could not be applied without known rainfall patterns. Thus, the historical rainfall patterns and the corresponding runoff hydrographs of all the 97 typhoon events were given to model the SFM, where 87 events for training and the remaining 10 events for testing. We noticed that because rainfall patterns were given, the results could only be treated as simulated rainfall-runoff patterns, rather than predicted runoff based on the previous rainfall histogram. Therefore, the simulation results were adopted only to assess the goodness-of-fit of our AI-based approach for predicting flood hydrographs. The results of the AI-based and SFM methods for the ten testing events are summarized in the Supplementary Table 1. It appears that the

proposed AI-based method improved the performance for all the test events, in terms of smaller RMS and larger R^2 values, especially apparent for events with high peaks. To demonstrate the goodness-of-fit of both methods, we examined performance for two special typhoon events, i.e. the most recent typhoon in 2019, Typhoon Lekima, and the highest flood hydrograph, Typhoon Aere in 2014 (Supplemental Figure 1). Results show that the AI-based approach is superior to the SFM method. This is especially true for Typhoon Aere, whose peak is the highest among those of 97 events. The results show that the AI-based approach with the best matched strategy could fit the historical flood hydrograph very well while the SFM method, which has three parameters calibrated based on training datasets, could not produce a suitable flood hydrograph and significantly underestimated the peak. Consequently, we conclude the AI-based approach can, in general, obtain reliable and accurate prediction of flood hydrographs based on known typhoon tracks and corresponding total rainfall amounts. (Lines 286-307)

From the AI perspective, while the AI tools seem to have been properly applied, the choice of AI tools would need to be justified via empirical comparisons to other methods used in the literature. For example, extreme storm track prediction has recently been done by a variety of supervised machine learning techniques, such as convolutional neural networks, using both storm track data and SST reanalysis data (wind, pressure) around the storm.

Response: Thank you for providing the valuable information regarding the supervised machine learning techniques, such as CNNs, for both storm track data and SST, which could be a great potential way to improve typhoon track prediction. This study focuses on flood hydrograph prediction based on given (predicted) typhoon track. We have added two references regarding the current development about tropical cyclone (TC) track forecasts, as shown below.

We notice that the quality of tropical cyclone (TC) track forecasts, especially for the track of the TC's center, has been significantly improved over the last three decades, where errors have been reduced by two-thirds in just 25 years²⁸, and an averaged error less than 100 km has been achieved for the test typhoon events²⁹. (Lines 330-333)

REVIEWERS' COMMENTS:

Reviewer #4 (Remarks to the Author):

This study presents an innovative way to forecast flow to the reservoir based on predicted typhoon tracks and self-organizing map (SOM), flow characteristic curve, and flood hydrograph.

The proposed approach can provide early warning for several days ahead of typhoon landfall; the process for flood prediction can be updated during typhoon period. The results show great potential for flood forecasting with long lead time. The manuscript is well written and will be of interest to readers of Nature Research. I recommend acceptance.

A few comments are listed below:

1. The proposed approach is based on historical typhoon tracks, rainfall pattern, and flow hydrograph. It is assumed that the typhoon tracks, rainfall, and flow are provided from historical record. However, the effectiveness of the method is highly relevant to its ability to predict future rainfall pattern, including the duration, intensity, and total amount. The rainfall pattern for each typhoon event can be unique. It is still a challenge for a numerical weather prediction model to make correct rainfall forecast for the next few days.

2. Flow characteristic curves (FCCs) are estimated using the total flow volume which is converted directly from total rainfall. The total rainfall for an event is not available before the typhoon event ended. The forecasted rainfall a few days before the event ended can vary greatly from the observed value. Similarly, the FCC is estimated based on the start rising limb characteristics and the cessation of rainfall. If the end time for the event is not forecasted correctly, the uncertainty of the forecasted flood hydrograph can be high.

3. Figure 3: the quality of the figure should be improved.

FINALLY: the title of the paper does not make sense to me. I Suggest revision, should be accepted. The definition of "Excavate...to..." may be unclear to readers of Nature Research.

Reviewer #4 (Remarks to the Author):

This study presents an innovative way to forecast flow to the reservoir based on predicted typhoon tracks and self-organizing map (SOM), flow characteristic curve, and flood hydrograph.

The proposed approach can provide early warning for several days ahead of typhoon landfall; the process for flood prediction can be updated during typhoon period. The results show great potential for flood forecasting with long lead time. The manuscript is well written and will be of interest to readers of Nature Research. I recommend acceptance.

A few comments are listed below:

Response: We sincerely thank you for recognizing our study. Your recommendation of acceptance for publication greatly encourages us. The insightful comments and constructive suggestions from you also enrich the final revision of our article. Thank you.

1. The proposed approach is based on historical typhoon tracks, rainfall pattern, and flow hydrograph. It is assumed that the typhoon tracks, rainfall, and flow are provided from historical record. However, the effectiveness of the method is highly relevant to its ability to predict future rainfall pattern, including the duration, intensity, and total amount. The rainfall pattern for each typhoon event can be unique. It is still a challenge for a numerical weather prediction model to make correct rainfall forecast for the next few days.

Response: Thank you for the insightful comments. Yes, a goal of this study was the prediction of flood hydrographs with a lead time of two days prior to typhoon landfall. The advantage of this approach is that the calculation of flow hydrographs no longer requires actual rainfall–runoff patterns that is only available post-event. As known, the rainfall pattern is the key element for modeling the rainfall-runoff process, while the whole rainfall pattern of a storm event could not be obtained in advance and the rainfall pattern could be varied widely. The rainfall pattern is not required in our methodology. The data set supporting the AI-based flood hydrograph prediction model developed in this research used total rainfall, typhoon track, the date and time warnings issued by our Central Weather Bureau (CWB) and hourly reservoir inflow data from our Water Resources Agency (WRA). To improve the accuracy and reliability of flood hydrograph prediction, our future study will incorporate the total rainfall amount and tropical cyclone (TC) velocity into our model.

2. *Flow characteristic curves (FCCs) are estimated using the total flow volume which is converted directly from total rainfall. The total rainfall for an event is not available before the typhoon event ended. The forecasted rainfall a few days before the event ended can vary greatly from the observed value. Similarly, the FCC is estimated based on the start rising limb characteristics and the cessation of rainfall. If the end time for the event is not forecasted correctly, the uncertainty of the forecasted flood hydrograph can be high.*

Response: Thank you for the valuable comments. Yes, hydrographs are commonly predicted from rainfall forecasts and typhoon duration. As known, traditional approaches (such as rainfall–runoff models) could only make short-term (hourly-based in our case) forecasts due to the lack of reliable rainfall pattern predictions. Therefore, it is imminent to improve traditional rainfall-runoff approaches and develop site-specific typhoon track–flood hydrograph prediction several days before typhoon landfall.

We argue that similar typhoon tracks would produce similar effects reducing analytical complexity. FCCs in the same cluster could be estimated by use of the total flow volume that was converted directly from total rainfall. We agree that *the forecasted total rainfall amount a few days before the event ended can vary greatly from the observed value by current techniques, which increase the uncertainty of the forecasted flood hydrograph.* We expect that typhoon track forecasts will be improved by the new prediction technology coupled with our CWB typhoon warnings before typhoon landfall. Thus, our model can be improved and accurate flood hydrograph predictions made by our approach will provide new critical information for flood defense and water management.

3. *Figure 3: the quality of the figure should be improved.*

Response: Thanks. We have re-plotted Figure 3, ensuring its resolution at 300 dpi.

FINALLY: the title of the paper does not make sense to me. I Suggest revision, should be accepted. The definition of “Excavate...to...” may be unclear to readers of Nature Research.

Response: Thanks for the insightful suggestion. We have revised the title as “Self-organizing maps allows typhoon tracks to make flood forecasts for up to two days”.